# Dual-Time-Point 18F-FDG PET/CT in Infective Endocarditis: Impact of Delayed Imaging in the Definitive Diagnosis of Endocarditis

**DOI:** 10.3390/biomedicines12040861

**Published:** 2024-04-13

**Authors:** Amanda Padilla Bermejo, Francisco José Pena Pardo, Edel Noriega-Álvarez, Mariano Amo-Salas, María de las Nieves Sicilia Pozo, Ana María García Vicente, Víctor Manuel Poblete-García

**Affiliations:** 1Nuclear Medicine Department, Ciudad Real General University Hospital, 13005 Ciudad Real, Spain; amanda.padilla.bermejo@gmail.com (A.P.B.); fjpena@msn.com (F.J.P.P.); abusici29@gmail.com (M.d.l.N.S.P.); vmanuelp@sescam.jccm.es (V.M.P.-G.); 2Nuclear Medicine Department, Guadalajara University Hospital, 19002 Guadalajara, Spain; 3Mathematics Department, Castilla La Mancha University, 13071 Ciudad Real, Spain; mariano.amo@uclm.es; 4Nuclear Medicine Department, Toledo University Hospital, 45007 Toledo, Spain; angarvice@yahoo.es

**Keywords:** infective endocarditis, dual-time-point, PET/CT, native valve, prosthetic valve

## Abstract

Infective endocarditis (IE) is a major public health condition due to the associated high morbidity and mortality. Our objective was to evaluate the utility of dual-time 2-deoxy-2-[18F] fluoro-D-glucose (18F-FDG) Positron Emission Tomography/Computed Tomography (PET/CT) imaging in the diagnosis of active IE in patients with suspected native valve endocarditis (NVE) and prosthetic valve endocarditis (PVE). For this purpose, a retrospective study was carried out, including patients suspicious of NVE or PVE who underwent a dual-time-point 18F-FDG PET/CT. A final diagnosis was established by the Endocarditis Team after patient follow-up using all the available findings. Sixty-nine patients were assessed. A final diagnosis of NVE was established in 3 patients of the 34 by 18F-FDG PET/CT and in the case of PVE was established in 20 patients of the 35. A statistically significant association was found when evaluating the association between PET diagnosis at early acquisition and final diagnosis of IE (χ^2^ = 30.198, *p* < 0.001) and PET diagnosis at delayed acquisition for final diagnosis of IE (χ^2^ = 9.412, *p* = 0.002). Delayed PET/CT imaging determined the IE diagnosis in 16/58 of the studies. In conclusion, delayed 18F-FDG PET/CT imaging seems to be useful in improving the definitive diagnosis of IE.

## 1. Introduction

Infective endocarditis (IE) is a major public health challenge [1]. In 2019, the estimated incidence of IE was 13.8 cases per 100,000 subjects per year, and IE accounted for 66,300 deaths worldwide [2]. Despite improvements in its management, it continues to be associated with high mortality and morbidity rates. Moreover, there is evidence of an increasing incidence. This may be due to the prolonged life expectancy of the population, as well as improved options for heart valve repair or replacement and/or enhancing the use of implantable cardiac electronic devices [3,4,5]. Early and accurate diagnosis is critical in IE and will have an important impact on the outcome. Abegaz et al. have recently published a systematic review and meta-analysis to examine, among other variables, the complications caused by infective endocarditis [6]. A total of 13,637 patients were evaluated. At least 1 complication was reported in 10,483 patients (76.9%), including renal, cardiac, and embolic (septic) complications.

The primary diagnosis of IE in most cases is based on a combination of clinical suspicion, imaging, laboratory criteria, and microbiological data. Bacteriae and other microorganisms such as viruses and fungi can cause IE, e.g., opportunistic oral bacteria, which can be identified by blood cultures [7]. The modified major Duke criteria were developed to facilitate the diagnosis of IE [8]. The modified Duke criteria have a sensitivity (se) and specificity (sp) of approximately 80% for native valve endocarditis (NVE) and significantly less for prosthetic valve endocarditis (PVE). Therefore, new imaging techniques are being developed to improve the diagnosis. Although echocardiography is the main diagnostic imaging technique used in IE, other non-invasive imaging techniques are increasingly used, including multislice computed tomography, magnetic resonance imaging, and nuclear imaging, i.e., 2-deoxy-2-[18F]fluoro-D-glucose Positron Emission Tomography/Computed Tomography (18F-FDG PET/CT) and leukocyte scintigraphy [9,10]. Furthermore, these imaging studies have been included as major diagnostic criteria in the modified Duke criteria in the European and American guidelines in 2015 [11,12] in the case of PVE, and as minor criteria in the recently published European and American guidelines in 2023 in the case of NVE [13].

In reference to diagnostic methods, Babes et al. recently published an article whose main objective was to assess the role of multimodality imaging in the diagnosis of IE [14]. Echocardiography is the imaging technique of choice and is the first performed as soon as IE is suspected [15]. Transthoracic echocardiography (TTE) is the initial method of investigation followed by transesophageal echocardiography (TEE) for further characterisation of lesions or identification of complications [13]. Three-dimensional (3D) echocardiography and intracardiac echography have an increased role in these situations [16,17]. The procedure of 3D echocardiography involves the utilization of a multiplanar probe that comprises a three-dimensional matrix array. The technique enables the evaluation of vegetations and valves in planes and angles that are not accessible with two-dimensional TEE [18]. The limitations of echocardiography are the difficulty in the evaluation of perivalvular complications, especially in PVE [19,20]. According to ESC guidelines, Cardiac Computed Tomography Angiography (CTA) is recommended as a class IB option for detecting valvular lesions and for identifying paravalvular and periprosthetic complications when echocardiography results are inconclusive. CTA demonstrates greater accuracy than TEE for assessing perivalvular and periprosthetic complications such as abscesses and pseudoaneurysms. However, TEE remains superior for detecting vegetations, leaflet perforation, and fistulae [13]. Further information is needed regarding the contribution of cardiac magnetic resonance imaging (CMR) to the diagnosis of IE [14]. CMR is able to show myocardial involvement in IE, detecting vegetations, and discerning the paravalvular extension of infection through delayed contrast enhancement [21,22]. 

18F-FDG PET/CT is a major additional tool in difficult cases of suspected IE. This method is particularly useful for identifying infections in various regions of the heart, especially in difficult cases such as those involving prosthetic valves, where TEE can be difficult despite its high se and sp values [23]. A prospective study involving 179 patients with suspected IE investigated the prognostic significance of 18F-FDG PET/CT. In patients with PVE, a significant correlation was found between a positive 18F-FDG PET/CT and adverse events such as unplanned cardiac surgery and death. A heightened FDG uptake was found to be associated with a higher frequency of embolic events in patients with NVE and PVE [24]. An additional major advantage of PET/CT is the possibility of whole-body imaging, rather than just the heart, to investigate for supportive findings to assist in making the diagnosis. Elevated splenic and bone marrow activity may serve as indirect indicators of IE. Consequently, there is a heightened glycolytic metabolism, manifested by augmented 18F-FDG uptake evident in both the spleen and bone marrow. A splenic-to-liver ratio greater than 1 has been proposed as supportive evidence for diagnosing IE [25,26]. Suspected accompanying spondylodiscitis and vertebral osteomyelitis can be evaluated by 18F-FDG PET/CT. This particular indication has received a Class I recommendation, supported by Level of Evidence C, according to the ESC 2023 guidelines [13]. The 18F-FDG PET/CT is not exempt from limitations. Among these, its limited value within the first 2 months following prosthetic valve implantation stands out, as FDG uptake may be present in the absence of any infection. Other limitations include its high cost, limited availability, radiation exposure, complex patient preparation, and the requirement for qualified personnel [14]. The added value of dual-time 18F-FDG PET/CT scans has been tested in various studies [27], both in oncology and in infectious and inflammatory processes. Mavi et al. [28] studied 152 patients with breast cancer who underwent dual-time-point 18F-FDG PET/CT PET imaging. 18F-FDG uptake increased over time in malignant neoplasia but decreased in normal breast tissue. Moreover, the procedural recommendations of cardiac PET/CT imaging, promoted by the European Association of Nuclear Imaging (EANM) and the European Association of Cardiovascular Imaging (EACVI), recommend a time interval between 18F-FDG administration and image acquisition of 60–90 min, but they also describe that a late PET acquisition might prove particularly useful in case of persistent high blood signal on PET images acquired 1 h post-injection [29]. Therefore, the majority of studies are performed under the “standard” protocol routinely used in oncology patients, performing chest imaging. Since the findings can be found in areas with high background activity, delayed PET/CT imaging could increase diagnostic accuracy by maximizing the contrast between the lesion and the background. For the clinical practice, adding focal tracer avidity around a prosthetic valve as a major criterion to the modified Duke criteria reduces the number of “possible” cases in a cohort of IE patients, and thus improves the diagnostic utility of these criteria [30]. For all these reasons, the aim of this retrospective study is to evaluate the efficacy of dual-time 18F-FDG PET/CT imaging in diagnosing active IE compared to standard methods in patients with suspected NVE or PVE. 

## 2. Materials and Methods

A retrospective study including consecutive patients with suspected NVE or PVE from April 2019 to January 2022 was performed.

### 2.1. Study Subjects

Patients referred by various clinical services, usually cardiology or medical internal medicine, to assess IE (in the vast majority of patients regardless of Duke’s classification) were included as a reflection of the daily clinical care practice. 

Patients who met the following inclusion criteria were included: (i) having undergone an optimized PET/CT imaging protocol consisting of dual-time-point acquisition (standard whole-body plus delayed thoracic scans), and (ii) clinical follow-up of at least 12 months after PET/CT. 

Different variables were collected, such as demographic data (age, sex), cardiological risk factors, immunosuppression risk factors (chronic obstructive pulmonary disease, chronic liver disease, smoking for more than 15 years, alcoholism), conditions causing immunosuppression (leukaemia, lymphoma, multiple myeloma, stage IV-V chronic kidney disease, diabetes mellitus, immunosuppressive/corticosteroid treatment, transplants, chemotherapy treatment in the previous 3 months, positive human immunodeficiency virus, rheumatologic diseases, autoimmune inflammatory conditions, and congenital immunodeficiencies), and the existence of previous infectious episodes. We also collected variables such as the presence of fever (> or <38 °C) at the beginning and during antibiotic treatment prior to the PET/CT scan, the patient’s previous history, as well as the date of implantation of the heart valve or implantable cardiac device, if any. In case of multiple surgical interventions to replace the same valve, the most recent implantation date was used.

Furthermore, data on blood cultures and the microorganisms involved were also collected and classified as positive if at least one of them was obtained before or during admission.

Echocardiographic findings, such as the presence of vegetations, abscesses, fistulas, and/or new dehiscence, and the involved valve were also recorded. If multiple echocardiograms were available, abnormalities in TEE were primarily considered.

Finally, based on these clinical features, the patients were evaluated before (pre-test) the 18F-FDG PET/CT scan according to modified Duke criteria and classified as rejected, definite, or possible IE [10,11,12].

### 2.2. PET/CT Acquisition and Imaging Evaluation

Patients followed a high-fat, low-carbohydrate diet for 48 h and fasted for at least 12 h before 18F-FDG administration to reduce physiological 18F-FDG uptake by the myocardium. Blood glucose levels obtained before 18F-FDG administration were classified as ≤160 mg/dL or >160 mg/dL.

A 3 MBq/kg dose of 18F-FDG was administered intravenously. PET/CT scans were performed using hybrid equipment (Discovery 5R-IQ GE HealthCare^®^, Chicago, IL, USA). A standard acquisition (from the skull base to the upper third of the lower extremities) was carried out 60 min after radiotracer administration, followed by a brain acquisition. A low-dose CT transmission study (modulated 120 kV and 80 mA) without enhancing contrast followed by a 3D emission PET scan was acquired (acquisition time of 2 and 5 min per bed for body and brain, respectively). A second set of images of the chest were acquired, usually at 150 min +/− 30 min (10 min/bed) after 18F-FDG administration. During the acquisitions, patients remained in a supine position with their arms raised above the head, except for brain acquisition.

PET images were reconstructed using CT ones for attenuation correction and after the application of an iterative reconstruction algorithm. 

Three experienced nuclear medicine physicians blindly and independently assessed the examinations by viewing both attenuation- and non-attenuation-corrected, PET, CT, and fusion images in axial, coronal, and sagittal views. Images of doubtful cases were evaluated jointly by the three nuclear physicians.

Myocardial uptake suppression was classified into two categories: complete inhibited uptake (less or equal to vascular pool uptake) or incomplete (diffuse or focal uptake superior to vascular pool uptake).

In case of incomplete myocardial suppression, it was assessed whether the images were interpretable or not.

IE criteria were qualitative and considered positive when any uptake of 18F-FDG higher than the background that met the features described below was visually detected. Globally, we can define the following [31]:–Uptake suggestive of infection: intense elevation of 18F-FDG (hypermetabolism) of focal and/or heterogeneous type in relation to prosthetic material or cardiac lesions, identified in both corrected and uncorrected images. In the case of PVE, it was also required that the prostheses had been placed more than 3 months ago.–Uptake not suggestive of infection: absence of uptake or mild homogeneous and/or diffuse 18F-FDG binding in relation to the prosthetic material.

A semi-quantitative analysis was also carried out to assess the intensity of 18F-FDG uptake, using the maximum standardized uptake value (SUVmax) in the abnormal area.

Late myocardial acquisition was also assessed both qualitatively and semi-quantitatively, comparing its results with early acquisition.

–Increased intensity in late acquisition compared to early acquisition was suggestive of infection.–Decreased intensity in late acquisition compared to early acquisition was suggestive of inflammation.–The sole appearance of increased focal or heterogeneous uptake in late acquisition was suggestive of infection.

### 2.3. Final Diagnosis and Diagnostic Impact of 18F-FDG PET/CT

The final diagnosis was established by the Endocarditis Team after patient follow-up of at least 4 months, using all the available results (clinical, microbiological, and imaging results, including 18F-FDG PET/CT).

The agreement between the 18F-FDG PET/CT standard scan and the final diagnosis was assessed. In the same way, agreement between delayed images and the final diagnosis was evaluated. The impact of delayed images in the final diagnosis of IE was also assessed.

Moreover, the impact of 18F-FDG PET/CT in raising or decreasing the Duke category was evaluated.

### 2.4. Statistical Analysis

Descriptive and statistical analysis was performed using IBM SPSS Statistics for Windows v.28 (IBM Corp., Armonk, NY, USA). In the descriptive analysis, categorical variables were described with absolute and relative frequencies. The association between categorical variables were studied with Pearson’s chi-square test (χ^2^ test). A *p*-value < 0.05 was considered statistically significant. 

The degree of agreement between techniques was studied with Cohen’s kappa coefficient (κ). A value of 1 would indicate a total degree of agreement. A multivariate logistic regression analysis was performed considering the final diagnosis by the multidisciplinary team as the dependent variable (gold standard). This analysis was carried out for patients with native and prosthetic valves independently, as well as a pooled analysis of them.

## 3. Results 

A total of 69 patients were analysed: 34 and 35 suspected of NVE or PVE, respectively, 42 men and 27 women, with a mean age of 65.77 ± 15.13 years (10–88). Patients and disease characteristics previous to 18F-FDG PET/CT are summarized in Table 1. 

Among all patients included, 84% were on antibiotic treatment, initiated an average of 11 days before the 18F-FDG PET/CT scan. 

Sensitivity, specificity, positive predictive value, and negative predictive value for echocardiography and 18F-FDG PET/CT in NVE and PVE are summarized in Table 2.

According to the gold standard, 7 (20.6%) of the patients were confirmed with NVE. A final diagnosis of NVE was established in 3 patients of the 34 by 18F-FDG PET/CT.

In the case of PVE, the gold standard classified 18 (51.4%) of the patients with confirmed PVE. A final diagnosis of PVE was established in 20 patients of the 35 by 18F-FDG PET/CT.

Of the 69 subjects included, 21 (30.4%) had inadequate myocardial suppression defined as persistent myocardial uptake greater than blood pool activity, but only 8 (11.6%) of them were not finally assessable; therefore, 61 patients were finally included in the study for the assessment of the contribution upon the use of late imaging.

Delayed imaging was performed in 58/69 (84%) subjects. Of those subjects, 17/69 (24.6%) had a positive interpretation in early images and 21/58 (36.2%) on delayed images (see Figure 1 and Figure 2). 

In total, the delayed imaging determined the diagnosis in 16/58 (27.6%) of the studies.

A statistically significant association was found when evaluating the association between PET diagnosis at early acquisition and final diagnosis of IE (χ^2^ = 30.198, *p* < 0.001) and PET diagnosis at delayed acquisition and final diagnosis of IE (χ^2^ = 9.412, *p* = 0.002).

When the degree of agreement was assessed, the following values were obtained: κ = 0.408, *p* < 0.001 for early acquisition, and a final diagnosis of IE and κ = 0.403, *p* = 0.002 for delayed acquisition and final diagnosis of IE.

In the multivariate analysis, only one result close to statistical significance (*p* = 0.052) was found. The OR value for fever in the prosthetic valve group was 9.273 (95% CI 0.97–87.86).

SUVmax values were also analysed. In the early acquisition, SUVmax values ranged from 2.8 to 13.3, with a mean of 4.85 ± 2.67 and in the late acquisition they ranged from 2.8 to 12.5, with a mean of 4.92 ± 2.60, showing an increase in SUVmax in all but one of the cases, which was also classified as positive for IE by visual assessment. 

Furthermore, integration of 18F-FDG PET/CT in the modified Duke criteria in PVE allowed appropriate reclassification of 8/17 patients from possible endocarditis to definite endocarditis category. 

## 4. Discussion

Infective endocarditis remains a diagnostic challenge due to its variable clinical presentation. The diagnosis is based on a clinical suspicion supported by consistent microbiological data and the documentation of IE-related cardiac lesions by imaging techniques [12].

As mentioned above, echocardiography, especially TEE, is the mainstay in diagnostic imaging of IE but is not free from some limitations. Cardiac CTA can overcome some of them [13,16,17,20]. 

Oliviera et al. published a systematic review and meta-analysis comparing the value of cardiac CT and TEE in IE. The sensitivity for detecting abscesses or pseudoaneurysms was higher with CT when compared to TEE, with values of 78% (95% CI: 70, 85%) versus 69% (95% CI: 62, 76%) (*p* = 0.052). The sensitivity for vegetation detection was significantly higher with TEE compared to CT, with respective values of 94% (95% CI: 92, 96%) versus 64% (95% CI: 57, 70%) (*p* < 0.001) [32].

Despite what has been explained, the help of functional imaging techniques is often required, especially in PVE, to support the diagnosis of IE. In this setting, 18F-FDG PET/CT has a significant role.

In a recent meta-analysis published by Wang et al., which included 26 studies involving 1358 patients, the following pooled sensitivities and specificities were estimated for PET/CT: se 0.31 (0.21–0.41, 29.4%) and sp 0.98 (0.95–0.99, 34.4%), and se 0.86 (0.81–0.89, 60.0%) and sp 0.84 (0.79–0.88, 75.2%) in NVE and PVE, respectively [23].

We analysed the potential added value of dual-time-point 18F-FDG PET/CT in the diagnostic workup of IE. The most important finding in our study was that delayed 18F-FDG PET/CT images could be useful to improve the diagnostic accuracy for IE both to confirm true positives and true negatives and to reclassify potential false positives or negatives since a statistically significant relationship between variables and correlations between techniques were observed for both hypotheses. Although it was not subject to analysis, it was visually observed that delayed images showed a higher contrast between target and background, due mostly to a decrease in blood pool activity.

After formulating the aim of this study, a literature search was carried out, revealing that there is a limited body of evidence on the subject.

Recent consensus guidelines on the role of 18F-FDG PET/CT in IE offered few evidence-based recommendations about the use of delayed imaging. The EANM in conjunction with the ECAVI published a guideline on recommendations for standardization of PET/CT imaging in inflammatory or infectious cardiovascular diseases, and, specifically regarding the delayed images in IE, it cites “120–180 min is sometimes applied to help assess inflammatory activity in the vascular wall and left ventricle due to lower background activity in the blood pool” [29].

But there is some evidence that delayed image acquisition could increase 18F-FDG-PET/CT diagnostic accuracy in suspected cardiac device-related infective endocarditis (CDRIE) when intracardiac lead infection is suspected [33,34] and in the diagnosis of NVE [35]. In the study by Abikhzer et al., in a subset of subjects, delayed imaging (134 ± 42 min) was conducted upon the request of the reading physician. This revealed that the target-to-background ratio (TBR) was higher on delayed images, with the increase in TBR correlating positively with the time interval between the 2 acquisitions (2.17 ± 0.48 vs. 1.65 ± 0.48, *p* < 0.0005). Moreover, delayed imaging identified a higher number of positive cases.

There are also two case reports in which late acquisition helped in the final diagnosis [36,37].

In addition, there is a bibliography with particular reference to the EANM Cardiovascular Committee document on PET imaging of atherosclerosis [38], recommending the acquisition of PET images 2 h after injection for reliable quantification of FDG uptake in the arterial vessel wall and/or plaques because delayed imaging presents the best compromise between a low background signal in blood and an acceptable duration of the PET study for patients.

The role of dual-time acquisition PET/CT in large-vessel vasculitis has also been studied by Martínez-Rodríguez et al. [39]. They concluded that the 180 min delayed 18F-FDG-PET/CT acquisition provides a more detailed visualization of the vessel wall, showing the washout of the blood pool activity. Another study [40] reported that the blood pool FDG activity was greater at one hour and decreased by 9.6% at the two-hour imaging acquisition time. A total of 10 subjects underwent 18F-PET/CT imaging at 65 min (95% CI 62 to 68 min) and 184 min (95% CI 181 to 187 min) after 18F-FDG administration. Blood pool SUVMEAN significantly decreased with time (*p* < 0.0001), whereas TBRMAX and TBRMEAN significantly increased with time (*p* < 0.005 and *p* =  0.020, respectively).

On the contrary, Scholtens AM et al. [41] analysed 14 scans in 13 patients referred for 18F-FDG PET/CT for suspicion of PVE, performed at standard (60 min post-injection) and late (150 min post-injection) time points. They scored them based on visual interpretation and semi-quantitatively with SUVmax and a target-to-background ratio, and they reported that they cannot recommend the use of delayed FDG PET/CT imaging because of the risk of false positive interpretation. Therefore, late imaging should be interpreted with caution.

Moreover, it is crucial to take into account various factors, including previous antimicrobial treatment, the size of vegetation, and blood glucose levels, as they might influence the accuracy of PET/CT results. Instances of false negative results have been documented in cases where antimicrobial therapy was previously administered [42,43]. Antibiotic usage before imaging impacts the diagnostic accuracy of 18F-FDG PET/CT imaging in IE. The levels of both systemic and local inflammation decrease as antibiotic therapy continues, leading to the occurrence of false-negative results [44,45,46]. In this study, no statistically significant relationship was found between the duration of treatment prior to 18F-FDG PET/CT and the final diagnosis of IE.

The strengths of the current study are the high number of patients included with a relatively long follow-up period, which provided an overview of clinical and analytical follow-up assessment of the patient. The availability of blood culture and echocardiography data made it possible for a correct pre-test classification with the Duke criteria. 

Study results are limited by the retrospective and single-centre design. Other limitations would be the large proportion of subjects with inadequate suppression, despite the use of an appropriate suppression protocol. Our findings should be corroborated by a prospective study.

## 5. Conclusions

According to the results of this study, in cases of suspected IE in NVE or PVE, when early imaging is non-diagnostic or inconclusive, late imaging could potentially increase diagnostic accuracy. No relationship with classical factors such as antibiotic treatment duration was found. These results need to be corroborated in a prospective study setting.

## Figures and Tables

**Figure 1 biomedicines-12-00861-f001:**
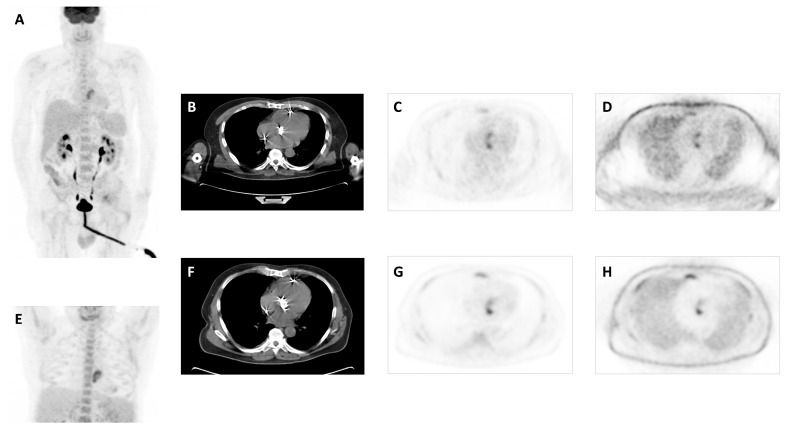
A 61-year-old male patient with a bioprosthetic aortic valve due to severe mitral insufficiency and stenosis with high fever for 17 days. Transthoracic and transesophageal echocardiography showed no paravalvular leakage or any thrombus on the valve leaflets or heart chambers. Two sets of blood cultures were negative. Patient with antibiotic treatment 12 days before PET/CT. MIP image (**A**), transaxial CT (**B**), attenuation-corrected PET (**C**), and non-attenuation-corrected PET images (**D**) showed focal and heterogenous increased 18F-FDG uptake at the site of aortic valve with 5.7 SUVmax, which increased to 6.9 on delayed images (**E**–**H**). Patient was finally diagnosed with confirmed PVE after PET/CT and completed a long-term treatment with antibiotics.

**Figure 2 biomedicines-12-00861-f002:**
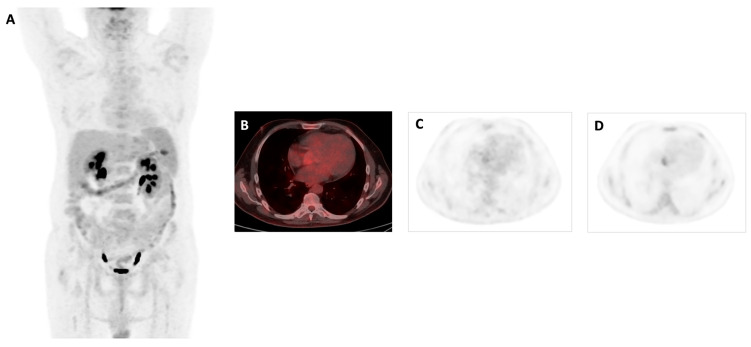
**A** 61-year-old male patient with no story of heart disease presented to the hospital with a high fever (up to 40 °C) and shivers daily for a month. In two sets of blood cultures grew *Enterococus faecalis.* Transthoracic echocardiography revealed 16 × 7 mm vegetation on the aortic valve. Antibiotic treatment started 7 days before PET/CT. 18F-FDG PET/CT images MIP (**A**) and axial views in fused (**B**) and PET (**C**) showed a faint focal uptake at the site of the aortic native valve, which increased on delayed imaging with a SUVmax of 4 (**D**). When we evaluated the impact of late imaging on the definitive diagnosis of IE, in the positive cases the late imaging showed an increase in intensity in both visual and quantitative assessment, except in one case, which was also assessed as visually positive. In the late acquisition, 6 new foci were observed on the late imaging and these findings were determined to be positive for active IE.

**Table 1 biomedicines-12-00861-t001:** Patients’ baseline characteristics.

	NVE (*n* = 34)	PVE (*n* = 35)
Mean age	57.9	68.6
Sex		
Male	21 (61.7%)	21 (60%)
Female	13 (38.3%)	14 (40%)
Blood cultures		
Positive	21 (61.8%)	21 (60%)
Negative	11 (32.4%)	13 (37.1%)
NA	2 (5.9%)	1 (2.9%)
Echocardiography		
Positive	13 (38.2%)	17 (48.6%)
Negative	20 (58.8%)	18(51.4%)
NA	1 (2.9%)	0 (0%)
Antibiotic treatment		
Yes	28 (82.4%)	30 (85.7%)
No	6 (17.6%)	5 (14.3%)
Fever		
Yes	27 (79.4%)	28 (80%)
No	7 (20.6%)	7 (20%)
Cardiological risk factors		
Yes	14 (41.2%)	33 (94.2%)
No	20 (58.8%)	2 (5.8%)
Infective risk factors		
Yes	16 (47.1%)	15(42.8%)
No	18 (52.9%)	20 (57.2%)
Immunosuppression risk factors		
Yes	17 (50%)	16 (45.7%)
No	17 (50%)	19 (54.3%)
Pre-test Duke criteria		
Rejected	8 (23.5%)	10 (28.6%)
Definite	7 (20.6%)	8 (22.8%)
Possible	19 (55.9%)	17 (48.6%)
Blood glucose level (*)		
≤160 mg/dL	28 (82.4%)	33 (94.2%)
>160 mg/dL	6 (17.6%)	2 (5.8%)

NA: not available; NVE: native valve endocarditis; PVE: prosthetic valve endocarditis; (*) previous to 18F-FDG administration.

**Table 2 biomedicines-12-00861-t002:** Analyses of echocardiography and 18F-FDG PET/CT for diagnosis of infective endocarditis and its subtypes.

	Sensitivity	Specificity	VPP	VPN
Echocardiography NVE	71%	70%	38%	90%
Echocardiography PVE	66%	70%	70%	66%
18F-FDG PET/CT NVE early acquisition	16%	95%	50%	80%
18F-FDG PET/CT PVE early acquisition	64%	64%	37%	64%
18F-FDG PET/CT NVE delayed acquisition	20%	90%	33%	82%
18F-FDG PET/CT PVE delayed acquisition	82%	64%	70%	78%

NVE: native valve endocarditis; PVE: prosthetic valve endocarditis; VPP: positive predictive value; VPN: negative predictive value.

## Data Availability

Data are unavailable due to privacy or ethical restrictions in Spain.

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
