# Peer review of "Dual-Time-Point 18F-FDG PET/CT in Infective Endocarditis: Impact of Delayed Imaging in the Definitive Diagnosis of Endocarditis"

_biomedicines, 2024, doi:10.3390/biomedicines12040861_

Round 1

Reviewer 1 Report

Comments and Suggestions for Authors

An interesting, informative and well written manuscript.  The article has both clinical interest and merit.  The article addressed this topic in a well-organized manner.  From the presented data, the conclusions are consistent with the findings.  The references are appropriate.  There are some editing issues that the authors should consider and address.  The following are suggestions/comments regarding those issues.  Line 14, "... a major public health condition due to the ...".  Lines 23 & 24, "... at delayed acquisition for the final diagnosis of ...".  Line 25, "... of the studies.  In conclusion, delayed ...".  Line 40, "... to facilitate in the diagnosis ...".  Line 44, "Furthermore, these imaging studies have been included ...".  Line 46, "... in the case of prosthetic valve endocarditis (PVE), and as ...".  Line 47, "... 2023 in the case of native valve endocarditis (NVE) [11]."  Lines 50 & 51, "... patients with suspected NVE or PVE from April 2019 ...".  Line 84, "Blood glucose levels obtained before ...".  Line 175, "... antibiotic treatment, initiated from an average ...".  Line 185, "... of the patients were confirmed with NVE."  Line 192, "... of the contribution upon the use of late imaging."   Line 233, "... patients from possibly being labeled as definitively having endocarditis."  Line 255, "... images in IE, it cites ...".  Lines 260 & 261, "... in the diagnosis of NVE [16].  In the study by Abikhzer et al., delayed imaging ...".  Line 271, "... because delayed imaging presents the best ...".  Line 275, "... Martinez-Rodriguez et al. [20].  They concluded that ...".  Line 277, "...blood pool activity.  Another study [21] reported that the ...".  Line 284, "... for suspicion of PVE, performed at ...".  Line 285, "... time points.  They scored them based on ...".  Line 292, "... elevated blood glucose levels may impact the ...".  Line 301 & 302, "... and analytical follow-up assessment of the patients."  Line 303, "it possible for a correct pre-test ...".  Line 310, "... of this study, in cases of suspected ...".  Line 311, "... late imaging could potentially increase the diagnostic ...".  Line 312, "... factors such as antibiotic treatment duration ...".

Reviewer 2 Report

Comments and Suggestions for Authors

1. Briefly explain why dual-time 18F-FDG PET/CT imaging is considered a promising diagnostic tool for IE, especially for readers unfamiliar with the technology.

2. Clearly state the study's hypothesis or primary research question, such as evaluating the efficacy of dual-time 18F-FDG PET/CT imaging in diagnosing active IE compared to standard methods.

3. Include details about the criteria used for patient selection in the retrospective study to clarify applicability and limitations.

4. Specify the time points used for early and delayed PET/CT imaging to improve understanding of the methodology.

5. Incorporate performance metrics (sensitivity, specificity, positive predictive value, and negative predictive value) of dual-time PET/CT imaging in diagnosing IE to enhance the results section.

6. Mention how PET/CT imaging results compare to traditional diagnostic methods for IE to highlight the study's contribution to the field.

7. Expand on the significance of the findings within the broader context of IE diagnosis and how this method could potentially change clinical practice or lead to better patient outcomes.

8. Ensure the abstract is free from grammatical errors and typos for better readability.

Comments on the Quality of English Language

 Moderate editing of the English language required

Reviewer 3 Report

Comments and Suggestions for Authors

Dear Authors,

Your research paper on the role of dual-time-point 18F-FDG PET/CT in the diagnosis of infective endocarditis is interesting and relevant to cardiovascular research. However, there are a few issues that need to be addressed:

  1. The definition of abbreviations must appear when first encountered, both in the abstract and in the main text – this is required per the instructions for authors (lines 15, 46, 47).
  2. The introduction section needs to be expanded. Please provide more information about the role of 18F-FDG PET/CT in the diagnosis of infective endocarditis. For suggestions, you can refer to https://doi.org/10.3390/life14010054.
  3. In Tables 1 and 2, the title should be placed above the table and the explanations below it, as per the instructions for authors.
  4. Please explain the rationale for choosing the cut-off value of 160 mg/dL for blood glucose level (line 85).
  5. Definitions of the abbreviations used in lines 252 and 253 should be provided.

Thank you for your attention to these matters. Addressing these issues will improve the overall quality of your manuscript.

Best regards

Round 2

Reviewer 2 Report

Comments and Suggestions for Authors

The authors have addressed all the comments. 

Comments on the Quality of English Language

Minor editing of English language required

Author Response

Thank you very much for taking the time to review this manuscript again. Please find the detailed responses below and the corresponding revisions/corrections highlighted/in track changes in the re-submitted files.

Point-by-point response to Comments and Suggestions for Authors

-Minor editing of English language required

We have revised the text again and have highlighted in the text, using yellow highlighting, modified words in an attempt to improve the English.

Specifically words/phrases in lines: 40-42, 45-47, 76-78, 80, 84, 84, 86, 90, 97, 98, 104, 311, 385.
